# Day Case Local Anaesthetic Thoracoscopy: Experience from 2 District General Hospitals in the United Kingdom

**DOI:** 10.3390/medsci11010023

**Published:** 2023-03-15

**Authors:** Megan Turner, Felicity Craighead, Joseph Donald MacKenzie, Avinash Aujayeb

**Affiliations:** 1Respiratory Trainee, Victoria Hospital, Hayfield Rd, Kirkcaldy KY2 5AH, UK; 2Respiratory Consultant, Victoria Hospital, Hayfield Rd, Kirkcaldy KY2 5AH, UK; 3Respiratory Consultant, Department of Respiratory Medicine, Northumbria Healthcare NHS Trust, Northumbria Way, Cramlington NE23 6NZ, UK

**Keywords:** local anaesthetic thoracoscopy, medical thoracoscopy, indwelling pleural catheter

## Abstract

Background: Local anaesthetic thoracoscopy (LAT) can be a vital procedure for diagnosis of unexplained pleural effusions. Traditionally, poudrage for pleurodesis and insertion of a large bore drain necessitated admission. There has been a shift towards performing LAT as a day case procedure with indwelling pleural catheter (IPC) insertion. This was advocated during the COVID pandemic by the British Thoracic Society (BTS). To determine the feasibility of such pathways, continuous evaluations are required. Methods: All day case LAT procedures with IPC insertion, performed in theatre, were identified at two large district general hospitals (Northumbria HealthCare in the North East of England and Victoria Hospital, NHS Fife, in Scotland). Rapid pleurodesis with talc was not performed due to local staffing problems. All patients had their LAT in theatre under conscious sedation with a rigid scope. Demographics, clinical, radiological and histopathological characteristics and outcomes were collected. Results: 79 patients underwent day case LAT. The lung did not deflate, meaning biopsies were not enabled, in four of the patients. The mean age was 72 years (standard deviation 13). Fifty-five patients were male and twenty-four were female. The main diagnoses were lung cancers, mesotheliomas and fibrinous pleuritis with an overall diagnostic sensitivity of 93%. Other diagnoses were breast, tonsillar, unknown primary cancers and lymphomas. Seventy-three IPCs were simultaneously placed and, due to normal macroscopic appearances in two patients, two large bore drains were placed and removed within one hour of LAT termination. Sixty-six (88%) patients were discharged on the same day. Seven patients required admission: one for treatment of surgical emphysema, four because they lived alone, one for pain control and one for control of a cardiac arrythmia. Within 30 days, there were five IPC site infections with two resultant empyemas (9%), with no associated mortality. Two patients developed pneumonia requiring admission and one patient required admission for pain management. The median number of days for which the IPCs remained in situ was 78.5 days (IQR 95). The median length of stay (LoS) was 0 days (IQR 0). No patients required further interventions for pleural fluid management. Conclusions: Day case LAT with IPC insertion is feasible with this current set up, with a median stay of 0 days, and should be widely adopted. The health economics of preventing admission are considerable, as our previous analysis showed a median length of stay of 3.96 days, although we are not comparing matched cohorts.

## 1. Introduction

The Coronavirus disease (COVID-19) pandemic has been a catalyst for widespread changes in the delivery of healthcare [1]. For pleural services in the United Kingdom (UK), as part of its guidance for services during the pandemic, the British Thoracic Society advocated the provision of day case local anaesthetic thoracoscopy (LAT) with indwelling pleural catheter (IPC) insertion to avoid hospital admission. LAT has been performed for many years by respiratory physicians worldwide, and it has a very favourable safety profile with excellent diagnostic sensitivity [2,3]. Traditionally, talc poudrage for pleurodesis occurs via a large bore chest drain (the last survey of LAT conducted in the UK by D. de Fonseka et al. showed some variation in practice, but showed that talc poudrage was implemented in 92% of centres. This was performed at the end of the LAT procedure), and the patient is admitted with an average length of stay (LoS) of 3 or 4 days [4]. Overall, 84% of centres offered simultaneous IPC insertion. Chatterji et al. evaluated the use of IPCs with or without talc poudrage at the time of LAT in a single UK centre in 36 cases of a possible trapped lung, finding that IPC insertion was feasible and reduced the need for further intervention [5]. Day case LAT has been described by Depew et al. [6] in 52 patients in a tertiary centre from 2011 to 2013 and then by Psallidas et al. [7] in 202 patients in five centres between 2010 and 2015. These studies confirmed the feasibility of the procedure, but there are no details on concurrent IPC insertion. Foo et al. provided the most up-to-date evidence in 45 patients with combined LAT and IPC insertion, with 87% of patients discharged on the day itself and a median LoS of 0 days, with no procedure-related deaths or infection. Foo et al. performed talc poudrage through a simultaneous large bore drain (removed thereafter) and pursued aggressive drainage post operatively: days to IPC removal had a median of 20 (IQR 13–48) days [8].

To continue providing LAT, Northumbria Healthcare NHS Foundation Trust, Newcastle, UK and Victoria Hospital, NHS Fife, Kirkcaldy, UK, started simultaneous IPC insertions at the time of LAT. Talc poudrage was deemed infeasible on both sites due to lack of space for the patients after the procedures. We hypothesised that simultaneous IPC insertions at the time of LAT, (day case thoracoscopy) was feasible and that same-day discharge was feasible without an increase in complications. 

## 2. Methods

A quality improvement project was registered with Northumbria Healthcare NHS Trust [NHCT] (QIP 623) and Caldicott clearance for the sharing of anonymised data was granted by Northumbria Healthcare NHS Trust (Reference WO78515). Informed consent was not required due to the retrospective nature of the study. The notes of consecutive patients undergoing day case LAT at Victoria Hospital, NHS Fife and NHCT between March 2020 and June 2022 were analysed. Demographics, clinical, radiological and histopathological characteristics and outcomes were collected. Continuous variables are presented as mean (±standard deviation) and categorical variables as percentages where appropriate. Feasibility was defined as being able to discharge more than 75% of patients on the same day. All analyses were performed in Excel, Microsoft 365, 2021. 

Patients for day case LAT were selected by the practising physicians: requirements for the procedure (diagnostic uncertainty and unexplained exudative pleural effusion as well as fitness) were not standardised. Both centres are high-volume pleural units and have experienced thoracoscopists. There is a similar set up (single 7-millimetre rigid thoracoscopy {Karl Storz Company, Tuttlingen, Germany}), which has been previously described. Pre-operative antibiotics are given on both sites. IPCs were to be routinely inserted, unless patient consent was not obtained, or if the operating physician deemed an IPC unnecessary (for example, if minimal fluid was present). The only difference between the sites is that in NHS Fife, LAT is performed with midazolam and fentanyl sedation, and that in NHCT, propofol and remifentanyl is used, alongside an erector spinae block (if a trained practitioner is available on the day). 

## 3. Results

Seventy-nine patients underwent day case LAT from the two centres (34 patients from NHS Fife, and 41 from NHCT). LAT requires lung deflation for inspection of parietal surfaces, but this was not possible in four patients due to significant adhesions in the pleural space. Seventy-five patients were thus analysed. The mean age was 72 years (standard deviation 13). Fifty-five (73%) of the patients were male and twenty-four (27%) were female. The diagnoses established are shown in Table 1. 

Four of the patients with chronic fibrinous pleuritis underwent video-assisted thoracoscopic surgery (VATS) and were diagnosed with mesothelioma thereafter. One patient was deemed unfit for VATS and was labelled as having radiological lung cancer. Thus, in 70 of cases, LAT provided the diagnosis, giving a sensitivity of 93%. Seventy-three IPCs were simultaneously placed and in two patients (due to normal macroscopic appearance and no concerning features for malignancy on contemporary imaging), two large bore drains were placed and removed post-operatively. Sixty-six patients (88%) who underwent day case LAT with IPC placement were discharged on the same day. Seven patients required admission. The reasons are listed in Table 2: 

Within 30 days, there were five IPC site infections with two resultant empyemas (9%) with no associated mortality. Two patients developed community-acquired pneumonias requiring admission and one patient required admission for pain management related to their tumour. The mean length of stay (LoS) was 0 days (standard deviation 0.2). No patients required further interventions for pleural fluid management. 

## 4. Discussion

This quality-improvement project shows that day case LAT with IPC placement is feasible, with just under 90% of patients discharged on the same day. We cover a very large geographical area and often ask if relatives can stay with the patients after their procedures. Sometimes, that cannot be achieved and patients have an unavoidable overnight stay. If we remove the 4 patients who lived alone from the dataset, then only 3 patients out of 75 required admission (4%). Previous data from NHCT [3] showed a median LoS of 3.96 days, which is now zero. Diagnostic sensitivity and complication rates across both sites are comparable to known evidence. Day case LAT with IPC placement is now routine practice in these two trusts. Patients with malignant pleural effusion also have shortened survival [9] and any reduction in their hospital stay might be important. Our study is from two centres and is larger and has some important differences. 

As far as we know, the only UK-centric case series published is from Foo et al. and has 45 patients. There are no published randomised controlled trials looking at day case LAT with IPC placement. Currently, in the UK, the Randomised Thoracoscopic Talc Poudrage + Indwelling Pleural Catheters versus Thoracoscopic Talc Poudrage only in Malignant Pleural Effusion (R-TACTIC) trial is recruiting [10]. The objectives are to determine the effect of combined LAT and IPC placement and to qualitatively assess the treatments in terms of their impact on patients and carers. The trial might reduce hospital stay and healthcare costs, improve quality of life and prevent further pleural procedures [10]. We already showed a reduction in LoS, although our comparative groups are not matched, from a previous assessment of the service in a single centre. However, formal health economics and patient-related outcome measures are important outcomes that need to be determined. In the R-TACTIC trial and in the case series by Foo et al., aggressive drainage via the IPC is performed. Aggressive drainage via IPC has been showed to increase the chance of pleurodesis in large randomised trials. The first one by Bhatnagar et al. [11], the IPC-Plus trial, showed a significantly higher rate of pleurodesis in 35 patients without significant trapped lung after talc instillation. In AMPLE-2 (Aggressive versus symptom-guided drainage of malignant pleural effusion via indwelling pleural catheters), daily drainage increased the pleurodesis rates at 60 days [12] and similar findings were replicated in ASAP [13] (Impact of Aggressive versus Standard Drainage Regimen Using a Long-Term Indwelling Pleural Catheter). Due to lack of resources for instillation of talc within 5 to 7 days of IPC placement, as well as lack of community nurses who would perform almost daily drainages, NHS Fife and NHCT do not perform these procedures. This, in turn, increases the amount of contact the patients have with the community drainage team and explains our median length of time of nearly 80 days for the IPC [14]. Whilst we have not measured the amount of nursing contact with the patients, Asciak et al. showed (notwithstanding an infection rate between 8 and 10%) that whilst IPCs reduce the number of subsequent pleural procedures, those patients visit hospitals for review more frequently and have a large number of planned visits for drainages. The infection risk might be theoretically increased by continued drain manipulation. We do not have data on how many visits the study patients have had. Our complication rate is 9% overall, which is similar to previous single-centre series [3] and to what is described in the British Thoracic Society statement on pleural procedures [15]. 

Out study has multiple limitations. This is a non-randomised observational project. There is no matched control group to compare day case LAT to and no formal health economics are possible. We also did not pursue aggressive drainage to encourage faster pleurodesis, so the median number of days the IPCs remained in situ was nearly 80 days. We did not count the number of the times the district nurses had to visit the patient to effect drainages and we did not calculate the cumulative cost of that. The COVID pandemic forced us to change our approach and provide day case thoracoscopy, and having lost inpatient beds with a significant service reconfiguration, we could not participate in the randomised trial. Another deviation from clinical practice is not inserting talc at thoracsocopy: we currently do not have an appropriate clinical space where we can observe patients post talc insufflation for any issues, and due to lack of community staff, as explained above, we cannot perform aggressive drainage in the first week to allow to talc insertion via the IPC in the days following the procedure. We exhibited a clear selection bias, as we pre-selected/pre-assessed patients who could undergo the procedure as a day case. Those who were not fit for day case procedures underwent image-guided (ultrasound) pleural biopsy and simultaneous insertion of an IPC: this was assessed via a randomised trial after the recent presentation of some semi-structured qualitative interviews [16]. Further work should also include the evaluation of the Clinical Frailty Scores or the Charlson Comorbidity Index in outcomes of LAT. 

In spite of the above limitations, whilst awaiting formal randomised evidence, day case LAT with IPC insertion can be performed with reduction in length of stay and preserve diagnostic sensitivity. Complication rates and health economic savings (if any) need to be studied in much larger datasets, with shared learning in between thoracoscopy practitioners to improve clinical outcomes. 

## Figures and Tables

**Table 1 medsci-11-00023-t001:** Diagnoses from thoracoscopic biopsies.

Diagnoses	Number of Cases
Lung cancer	22 (29%)
Mesothelioma	18 (24%)
Breast cancer	2 (3%)
Chronic fibrinous pleuritis	28 (37%)
Cancer of unknown primary	2 (3%)
Lymphoma	2 (3%)
Tonsillar cancer	1(2%)

**Table 2 medsci-11-00023-t002:** Reasons for admission post thoracoscopy and length of stay (days).

Reason for Admission	Number of Cases	Length of Stay (Total Days)
Patient lived alone	4	4
Pain control	1	1
Control of cardiac arrythmia	1	1
Progressive surgical emphysema	1	1

## Data Availability

Data can be shared upon reasonable requests.

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
