# Peer review of "Day Case Local Anaesthetic Thoracoscopy: Experience from 2 District General Hospitals in the United Kingdom"

_medsci, 2023, doi:10.3390/medsci11010023_

Round 1

Reviewer 1 Report (Previous Reviewer 1)

Dear Editor and Authors,

Thank you for asking me to re-review this work and provide my opinion once more. Having read both the new manuscript and also the response of the authors I must say that although I was slightly negatively predisposed towards this work initially because I felt it represented a presentation of a “stop gap” solution during a difficult period which we shouldn’t really advertise, I have now changed my opinion!!

The work still has significant inherent biases and problems but the authors have followed my suggestions and have presented and explained the reasons behind them quite well! This in my eyes strengthens the paper since it now represents as mentioned above an effort to continue providing a service to patients during a difficult period with minimal support and experience!

Therefore, I am happy to suggest the publication of the work.

Good luck to all.

Reviewer 2 Report (Previous Reviewer 2)

Reviewed

Reviewer 3 Report (Previous Reviewer 4)

Well done.

This manuscript is a resubmission of an earlier submission. The following is a list of the peer review reports and author responses from that submission.

Round 1

Reviewer 1 Report

Dear Editor and Authors,

Thank you for asking me to review this work titled “Day Case Local Anaesthetic Thoracoscopy: Experience from 2 District General Hospitals in the United Kingdom” in which the authors present their experience with day surgery pleuroscopy in two hospitals in the large DGIs in the United Kingdom. The authors had to rely on this approach due to the restrictions imposed by the pandemic. In this manuscript therefore they present their cohort outcomes of 79 patients.

However, I must say this is not something new at all!! As a junior thoracic surgery registrar at King’s we were performing thoracoscopic day surgery since 2007 particularly for such cases as reported by the authors!

In addition, the avoidance of performing talc poudrage is a significant limitation and although it might have been dictated by the pandemic circumstances represents a significant deviation from acceptable clinical practice!!

Moreover, there is significant selection bias as even admitted by the authors as only fit and able to be discharged patients - of course this means ambulatory and in good performance status were able to undergo day procedures.

The number/percentage of admitted patients 7/75 = 9.3% is quite high and more likely represents poor patient selection (as evident that 4 patients lived alone and could not be discharged – this is prior info well known in advance).

In addition, the number/percentage of complications, 9% wound infection for example and 2 empyemas is quite high!! This more likely represents unfamiliarity with day case surgery and protocols (which as the authors themselves admit had to quickly implement due to the pandemic) and are results of the learning curve we all face.

In conclusion, I do not feel the presentation of this audit (for this is what it is essentially, a quality improvement project) has much to offer the clinical community! It shows what we were forced to do during the pandemic to deliver any sort of semblance of healthcare to our patients but I would not really boast about it!! If anything, this work would be more acceptable if its focus and message was more on “this is what we had to do, it wasn’t the best this is why our results are poor so we need to see how next time we can do better and offer suggestions/tips the team learned from the experience”. I am really sorry, but I can’t support its publication at its present form. I wish all well.

Reviewer 2 Report

The methods indicate that the mean and standard deviation were used for continuous variables (line 76). However, they then present medians.

On line 22 of the summary, the mean is presented with its range. The authors must verify if the distribution is normal for age, record the mean with its standard deviation as established by the methodology, otherwise present the median and its respective range. The same thing happens on line 95.

Table 1. In the methodology they indicated that frequencies would be used for the categorical variables, however, they only present absolute numbers.

Table 2. The presentation of the information should be complemented or improved. Table two has no major relevance as it is at the moment

Describe where the analyzes were performed (software)

Reviewer 3 Report

This paper is a reasonably presented audit of everyday practice. It once again confirms the feasibility to perform thoracoscopy under local anaesthetic and demonstrated the limitations of the procedure in case of complications or the necessity to expand the procedure.

Author Response

Thank you- no specific comments are required with this

Reviewer 4 Report

Nice work.

you should have added Charleston Morbidity Index or something else dealing with frailty - I would not leave sombody alone over night after a medical thoracoscopy.

For pleurodesis you have to remember that renal insufficiency is an own and predominant risk factor as this can occur after talc application.

Author Response

This has been added, thank you